



# Characterization of Dynamic Stall on Large Wind Turbines

Hye Rim Kim, Jasson A. Printezis, Jan Dominik Ahrens, Joerg R. Seume, and Lars Wein

Institute of Turbomachinery and Fluid Dynamics, Leibniz University Hannover, Germany

**Correspondence:** kim@tfd.uni-hannover.de

**Abstract.** This study shows an extensive analysis of dynamic stall on wind turbine airfoils preparing the development of a reduced-order model applicable to thick airfoils ($t/c > 0.21$) in the future. Utilizing URANS simulations of a pitching FFA-W3-211 airfoil at the Reynolds number of 15 million, our analysis identifies the distinct phases in the course of the evolution of dynamic stall. When the dynamic stall is conventionally categorized into the primary
instability transitioning to the vortex formation stage, we suggest two sub-categories in the first phase, and an intermediate stage featuring a plateau in lift prior to entering the full stall region. This delays the inception of deep stall, approximately 3° for a simulation case. This is not predictable with existing dynamic stall models, optimized for low Reynolds number applications. These features are attributed to the enhanced flow attachment near the leading-edge, restricting the stall region downstream of the position of maximum thickness. The analysis on the frequency
spectra of unsteady pressure confirms the distinct characteristics of the leading-edge vortex street and its interaction with large-scale mid-chord vortices to form the dynamic stall vortices (DSVs). Examination of the leading-edge suction parameter (LESP) proposed by Ramesh et al. (2014) for thin airfoils under low Reynolds numbers reveals that LESP is a valid criterion in predicting the onset of the static stall for thick airfoils under high Reynolds numbers. Based on the localized separation behavior during a dynamic stall cycle, we suggest a mid-chord suction parameter
(MCSP) and trailing-edge suction parameter (TESP) as supplementary criteria for the identification of each stage. The MCSP exhibits a breakdown in magnitude at the onset of the dynamic stall formation stage and full stall, while TESP supports indicating the emergence of a deep stall by detecting the trailing-edge vortex.

## 1   Introduction

Power generation of wind turbines (WTs) increases the square of their rotor diameter, driving the trend towards
larger WTs. The next generation of offshore megastructures will reach rated capacities of approx. 20 MW and diameters of 350 m. The elongated and flexible rotor blades of these megastructures are more prone to deformations, which along with wind speed fluctuations, turbulence, and altitude-dependent wind distribution, locally alter the blade's angle of attack (AoA). If the AoA exceeds the static stall angle, it can trigger dynamic stall. Dynamic stall introduces transient loads that potentially excite blade vibrations, which contribute to mechanical fatigue and can
lead to blade failure. It is, therefore, crucial to predict the onset of dynamic stall and account for the increased dynamic loads in the design of future WT blades. Dynamic stall has long been a subject of research in helicopter





aerodynamics (Leishman, 2006; Corke and Thomas, 2015), it has been investigated both experimentally (Merz et al., 2017; Schwermer et al., 2019), and numerically (Letzgus et al., 2019). Differences between dynamic stall on helicopter and WT blades are the larger diameters and chord lengths, resulting in higher Reynolds numbers (Re $\approx$ 15 M) and

lower Mach numbers (Ma < 0.3). Additionally, WT airfoils have a higher thickness-to-chord ratio of $t/c > 0.21$ and Bangga et al. (2017) has shown that the flow around WT blades can be assumed quasi-three-dimensional (Q3D) for regions outside of the inner 30% of the span. Sharma and Visbal (2019) investigated the influence of airfoil thickness on dynamic stall and found that the influence of trailing-edge separation increases with thickness. They noted that for Q3D simulations, a span of 10% of the chord is sufficient to study the onset of dynamic stall. The stall behaviour

of an airfoil gradually shifts from trailing-edge stall to leading-edge stall, and the stall angle and maximum lift increase for growing Reynolds numbers above 2 million (Brunner et al., 2021). Kiefer et al. (2022) found that for thick airfoils, the stall delay is characterized by a power law that is a function of the Reynolds number, kinematics of the pitching motion, and airfoil geometry parameters. Huang et al. (2020) identified the freestream's turbulence as a critical parameter that delays the onset of dynamic stall. The higher the freestream's turbulence, the later

the dynamic stall vortex (DSV) forms because the turbulence increases the transport of momentum in wall normal direction and, thus, stabilizes the boundary layer. The three stages of dynamic stall according to Mulleners et al. (2012) are the primary instability stage, vortex formation stage, and deep stall. When exceeding the static stall angle $\alpha_{\text{ss}}$, the boundary layer enters the primary instability stage, which is characterized by the formation of small eddies on the suction side of the airfoil and an increase in lift above the maximum of the static curve. Further increasing the

angle of attack above $\alpha_*$ initiates the vortex formation stage. The unstable boundary layer rolls up and the DSV is formed. This vortex subsequently detaches and the airfoil reaches deep stall, which leads to a sudden drop in the lift. Decreasing the angle of attack prior to exceeding $\alpha_*$, prevents the formation of a DSV and keeps the airfoil in the light stall regime, which is associated with significantly lower dynamic blade loads and load oscillations (Mulleners et al., 2012; Deparday and Mulleners, 2019). The total stall delay is composed of the stall delay attributed to the

primary instability stage and the stall delay attributed to the vortex formation stage. The first is known to be a function of the airfoil geometry, whereas the later depends on the freestream's conditions and kinematics of the airfoil (Mulleners et al., 2012; Kiefer et al., 2022).

The operating conditions of WTs are highly unsteady and vary over the span of the airfoils. Preventing the local angle of attack from exceeding the static stall angle at every span-wise position and, thereby, avoiding dynamic

stall is an impossible task with global pitch control. Changing the pitch angle locally would require expensive gear that measures the local angle of attack at multiple span-wise positions and an approach to locally reduce that angle of attack. Gerontakos and Lee (2006) and Andersen et al. (2009) investigated trailing-edge flaps as a way to avoid dynamic stall. However, installing such devices in a WT would drastically increase the cost and the maintenance intervals. A cost-effective way of mitigating the effects of dynamic stall is to develop airfoil geometries

that are inherently less prone to dynamic stall. In order to design such airfoils, the angle where the DSV forms, $\alpha_*$ has to be predicted and maximized for new airfoil designs. This is possible because $\alpha_*$ is a function of the airfoil





geometry and flow conditions. However, current dynamic stall models that are implemented in the blade element momentum (BEM) theory require empirical results and rely on parameters that are derived from the static lift curve of already existing airfoils. These lift curves need to be obtained via expensive experiments or CFD simulations and

are acquired for every combination of airfoil geometry and aerodynamic boundary conditions that are investigated in the BEM simulation. Applying these empirical results to new airfoil geometries leads to large uncertainties in the BEM simulations (Tangler, 2002; Simms et al., 2001). This demonstrates the need for a non-empirical ROM for predicting the onset of the vortex formation stage. An example for such an approach for thin airfoils under low Reynolds number was established by Ramesh et al. (2014), who introduced the critical leading-edge suction

parameter (LESP$_{\text{crit}}$). The LESP is a measure of the suction at the leading edge. It is calculated by integrating the local force at the airfoil surface in the interval $0 < x/c < 0.1$ and extracting the chord-wise component. They found that for any thin airfoil and Reynolds number, there exists a critical LESP$_{\text{crit}}$. When exceeded, vortex shedding occurs at the leading-edge marked by a sudden breakdown of the suction of the airfoil and its lift. The temporal evolution of the LESP can be predicted by models that calculate an unsteady pressure distribution around the airfoil

(e.g. unsteady vortex-lattice method (Konstadinopoulos et al., 1985)). LESP$_{\text{crit}}$ is a function of the airfoil geometry and Reynolds number and using thin airfoil theory, we can predict it with the first term of the Fourier series of the vortex sheet strength distribution along the camber line. Using this relationship, Deparday and Mulleners (2019) predicted the onset of the vortex formation stage for thin airfoils based on LESP$_{\text{crit}}$ and improved the method by introducing an effective angle of attack that depends on the instantaneous shear layer height at the suction side of

the airfoil. Mulleners et al. (2012), Gupta and Ansell (2019), and Sharma and Visbal (2019) state that for thick airfoils the DSV does not form at the leading-edge but at mid-chord. This was experimentally confirmed by Kiefer et al. (2022).

Based on these observations, the authors explore dynamic stall of a thick airfoil at a Reynolds number of 15 million. The vortex formation stages and deep stall are investigated, characterizing the local flow separations. Our hypothesis

is, that there exists the critical mid-chord suction parameter (MCSP) and trialing-edge suction parameters (TESP), which can be used to identify the different formation phases of the dynamic stall vortices. The MCSP is a measure of the suction at mid-chord in the interval $0.3 - 0.4$ $c$ directly downstream of the maximum thickness at $0.3$ $c$, and TESP in $0.9 - 1$ $c$. All of the LESP$_{\text{crit}}$, MCSP$_{\text{crit}}$, and TESP$_{\text{crit}}$ are criteria that predict vortex shedding along the airfoil surface without relying on empirical static polars as the BEM method. We test our hypothesis by conducting

URANS simulations of dynamic stall at a Reynolds number of 15 M. This Reynolds number will be reached by future offshore WTs and, to the best of our knowledge, dynamic stall has not been comprehensively investigated at such operating conditions.





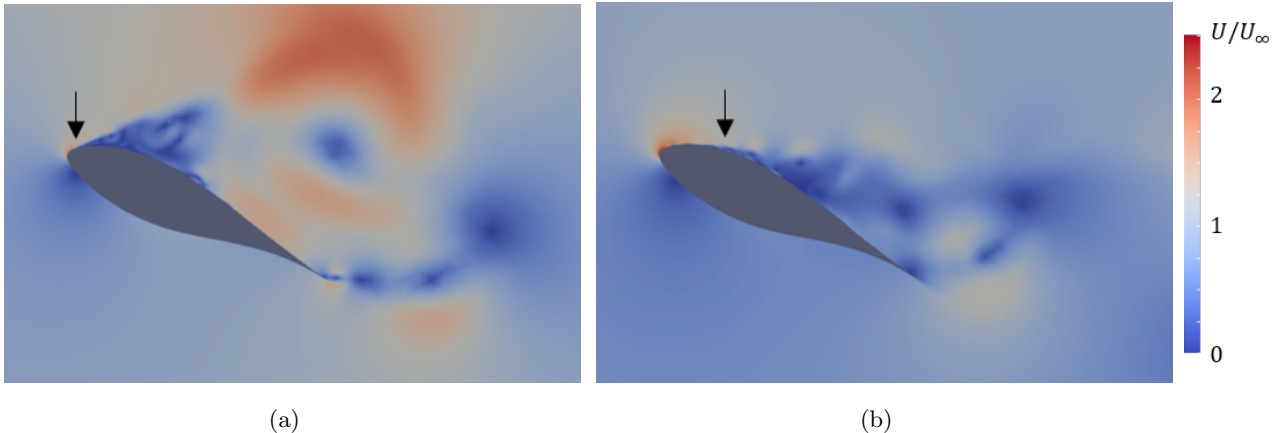

(a)                           (b)

**Figure 1.** Instantaneous contour of velocity during a pitching motion: (a) Re = 1.6 M, showing typical leading edge separation and large-scale dynamic stall vortex formation. (b) Re = 16 M, small-scale vorticies attached in leading-edge region, delaying the onset of deep stall

## 2 Methodology

### 2.1 Numerical Setup

OpenFOAM v2012, an open-source CFD software, is used to conduct a three-dimensional time-resolved RANS simulation for investigating dynamic stall on the FFA-W3-211 airfoil (Fig. 1). The FFA-W3-211 airfoil is a commonly used geometry for WTs with a thickness-to-chord ratio of 0.21 at $0.3\ c$ and the coordinates of the geometry can be found in Björck (1990). The main purpose is to predict the flow separation on the suction side of the airfoil for both, the static and the dynamic cases at a Reynolds number of 15 M, which is representative for future offshore WTs.

The computational domain and mesh are presented in Fig. 2. The Q3D domain consists of the stationary outer annulus with the diameter of $45\ c$ and the pitching inner cylinder with the diameter of $5\ c$, having the chord length of $c = 3.5$ m. The two domains are coupled at each pitching motion using cyclicAMI boundary condition in OpenFOAM. The span of the Q3D model is set as $2.5\ c$. Boundary layers at the blade surface are fully resolved, i.e. the non-dimensional grid spacing in wall normal direction $y^+$ is smaller than 1. The cell account in the spanwise

direction is 20, resulting in relatively high $z^+$ in the order of $10^3$. This is a compromise between the computational cost and resolving three-dimensional vortices. The influence of cell numbers in the spanwise direction is shown in the following section. This results in total number of cells of 0.7 Mil for the reference case. At the inlet, a uniform velocity of $U_\infty = 80$ m/s and a zero gradient for the kinematic pressure are applied, while a zero gradient for the velocity and a uniform kinematic pressure of $p/\rho = 0$ (incompressible flow) is applied at the outlet. The turbulent

quantities are imposed at the inlet as fixed values, which corresponds to the turbulence intensity of 0.01%. A periodic condition is used for all wall boundaries, except for the airfoil surface, where a zero-velocity condition is enforced.





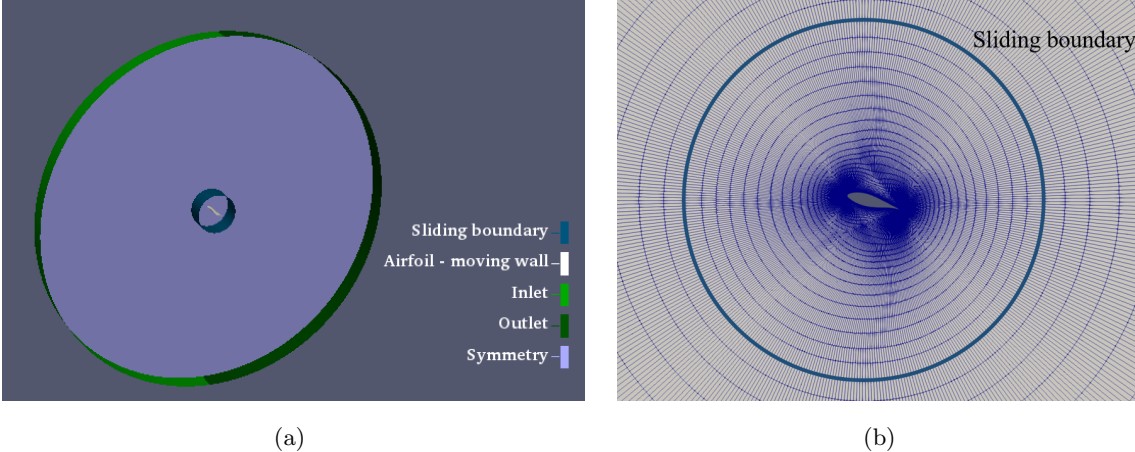

(a)                                                (b)

**Figure 2.** Numerical domain (a) boundary conditions (b) mesh around the airfoil, blue line indicating the interface between pitching and stationary domains

The transient and incompressible pimpleFoam solver has been used for the unsteady RANS simulations. The two-equation shear stress transport (k-$\omega$ SST) model is applied as the turbulence model. It uses the PIMPLE (merged PISO-SIMPLE) algorithm for the pressure and velocity coupling (Issa, 1986). The transition model is not applied in this study since the flow is mostly turbulent (Fig. 1b). The correction of dominant fluxes in the impacted cells considers the impact of mesh movement. This correction involves substituting the velocity with a relative velocity in all convection terms. A comprehensive explanation of this process can be found in the reference Jasak (2009). OpenFOAM uses the finite volume method to discretize the differential RANS equation. The temporal discretization of implicit $2^{nd}$ order, and the spatial discretization of $2^{nd}$ order Gauss linear schemes are employed with the exception of Laplacian schemes of Gauss linear limited corrected 0.5. This setup was successfully validated by Ahrens et al. (2022). The timestep is adjusted during the unsteady simulation with the maximum CFL number of 10 along with the maximum time step of $5e-4$ s. Although the maximum CFL number is set to be 10, local CFL number is equal or less than 1, except at the LE nose and TE where the mesh is excessively fine and velocity is high. This results in the time steps varying between $5e-5$ s at stable region, and $6e-6$ s at stall region within a cycle.

To evaluate the static and dynamic performance of the airfoil, 200 numerical probes are located along the surface at the mid-span, with a denser distribution near the leading-edge. The data is sampled at the frequency of 100 kHz for the spectral analysis, this is $100-1000$ times down-sampled for the quasi-steady state analysis. The total simulation time corresponds to 5-15 cycles, then the first one or two are excluded to present the fully developed cycle behavior.



## 2.2 Blade Element Momentum Code

The most common engineering method for predicting dynamic stall with low computational effort is by integrating a dynamic stall model into a blade element momentum (BEM) code. Detailed presentations of the implementation of dynamic stall models in the BEM theory for forecasting unsteady aerodynamic loads on WTs have been conducted in previous publications and are summarized here for reference (Branlard et al., 2022).

The BEM theory combines the momentum and the blade element theory to ascertain the aerodynamic characteristics and loads of WT rotor blades. The momentum theory characterizes flow dynamics within a control volume, allowing the calculation of thrust and torque. The blade element theory divides the blade into infinitesimal segments, enabling analysis of individual radial positions. By integrating these theories, BEM determines the forces and moments at different radial positions. However, the theory does not allow the determination of forces along the chord direction.

To increase the accuracy of BEM models, correction models have been developed over time. These correction models do not change the fundamental calculation algorithm of the BEM theory and thus are unable to calculate pressure forces over the chord length of an airfoil. However, the correction models can extend the BEM theory by taking into account additional physical effects, e.g. to consider unsteady aerodynamic effects like dynamic stall. The most widely applied dynamic stall model in literature for wind turbines is the BLM model (Leishman and Beddoes, 1989). The BLM model consists of four modules are used to calculate dynamic stall effects, which are then linearly combined to obtain the resulting unsteady lift. More information about these modules can be found in Gupta and Leishman (2006). The foundation of the BLM model rests upon calculated static polar information. According to Simms et al. (2001) and Tangler (2002) the largest source of error in rotor load and performance predictions is suspected to be due to incorrect static polar information. Thus, the quality of a dynamic stall model is largely dependent on the quality of the static polar data calculated. The static polar information for a specific airfoil can be determined either experimentally or by CFD simulations.

## 2.3 Test Cases

The URANS simulation is first conducted at fixed AoA as listed in Tab. 1 to attain static polars. This is compared to XFOIL and utilized as an input of BEM calculations. The dynamic cases are simulated by pitching AoA, $\alpha$, defined by

$$\alpha(t) = \bar{\alpha} - \hat{\alpha} \cdot cos(2\pi f_{\mathrm{p}} t) \tag{1}$$

in Tab. 1. The pitching frequency is $f_{\mathrm{p}} = 1$ Hz, which corresponds to a reduced frequency of $k = 0.137$. Simulations at Reynolds numbers of 1.6 M and 16 M (e.g. Fig. 1) and $k$ in a range of $0.029 - 0.137$ have been conducted, too. However, the conclusions regarding the formation of dynamic stall on future large WTs remain the same, which is why the results are not shown in this paper.





**Table 1.** Test cases at Reynolds number of 15 million

| Case ID | | mean AoA, $\bar{\alpha}$ in ° | pitching angle, $\hat{\alpha}$ in ° | k |
|---|---|---|---|---|
| 1 | Static | 14, 17, 20, 23, 26, 29, 32, 35 | - | - |
| 2 | Dynamic | 17 | 8, 15 | 0.137 |
| 3 | Dynamic | 20 | 5, 10, 15 | 0.137 |

## 2.4 Non-dimensionalization

Comparability of the results presented in this paper is ensured by providing non-dimensional flow quantities. The non-dimensional quantities analyzed in this work are listed in the following paragraph. The Reynolds number

$$\text{Re} = \frac{c \cdot U_\infty}{\nu} \tag{2}$$

is based on the chord length $c$, the freestream velocity $U_\infty$, and the kinematic viscosity $\nu$. A local Reynolds number, $\text{Re}_x$ takes axial position $x$ instead of the chord length $c$. Strouhal number St is given to describe characteristic frequency of the vortex shedding $f$,

$$\text{St} = \frac{f \cdot c}{U_\infty} \tag{3}$$

The kinematic frequency, in our case the pitching frequency $f_\text{p}$, is non-dimensionalized as reduced frequency $k$ following the convention:

$$k = \frac{\pi \cdot f_p \cdot c}{U_\infty} \tag{4}$$

The non-dimensional cell height at the wall

$$y^+ = \frac{y \cdot \sqrt{\tau_\text{w}/\rho}}{\nu} \tag{5}$$

is calculated with the cell height at the wall $y$, the wall shear stress $\tau_\text{w}$ and the density $\rho$. $x^+$ uses the stream-wise cell size as input and $z^+$ the span-wise size, respectively. The lift coefficient is based on the lift force of the airfoil $l$:

$$c_\text{l} = \frac{l}{\rho U_\infty^2 c} \tag{6}$$

The drag coefficient $c_\text{d}$ is the non-dimensionalized drag force $d$ in the same manner.

## 3   Mesh and Time-Step Studies

The preliminary studies are discussed in this section prior to the main analysis. The static polar is calculated by averaging the pressure after the convergence of the non-pitching simulations. The static case is utilized as the



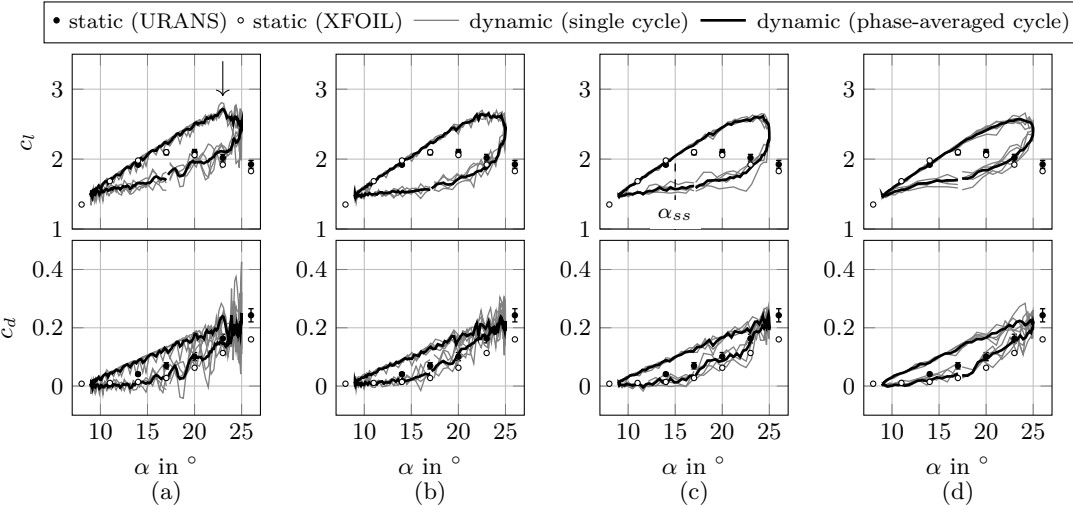

**Figure 3.** $17°\pm8°$ pitching case for mesh study of different cell counts in span (a) 1 cell (b) 10 cells (c) 20 cells (reference) for the airfoil aspect ratio of 2.5, (d) 20 cells for the airfoil aspect ratio of 1.25. Error bar of static polar indicates 95% confidence level of URANS simulation.

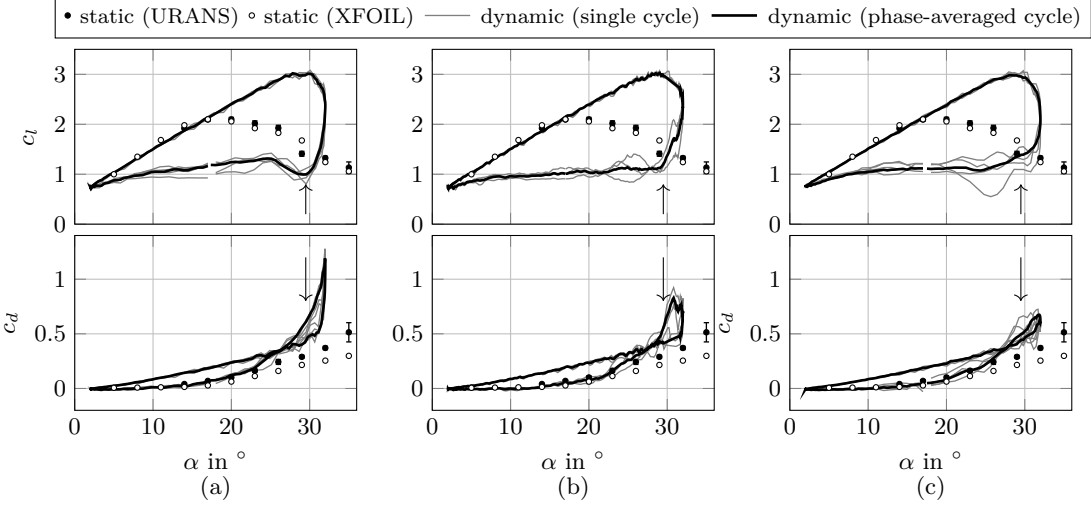

**Figure 4.** $17°\pm15°$ pitching case for time-step study of maximum CFL number in (a) 100 (b) 50 (c) 10 (reference).

initial solution for the pitching cases. The dynamic cases have been simulated for 5-15 cycles, the data is then phased-averaged after the convergence. Since the time increment varies at each time-step in unsteady setups, the data is resampled at a constant sampling frequency. Most of the contour figures are based on this phase-averaged data, e.g., the spatiotemporal and spatiospectral plots, except a few instantaneous plots. Besides the phase-averaged






performance, it is occasionally shown together with each cycle to represent the cycle-to-cycle variations. In this section, a maximum 4 cycles are shown from each case for the simplicity, even though the phase-average cycle is a result of averaging 5-15 cycles.

The mesh study and time-step study have been conducted to find the adequate steps of spatial and temporal discretizations. Three cases of different cell counts of 1, 10, and 20 in the span of the aspect ratio (AR) of 2.5 are simulated with the pitching angle of $17° \pm 8°$. The other case is AR of 1.25 with 20 cells in the span. Figure 3 shows the dynamic stall cycles from four mesh setups together with two static polars, one from time-averaged URANS of non-pitching setups of 20 cells in the span, the other from two-dimensional simulations in XFOIL. The error bar indicates the confidence level of 95% of URANS simulations, which enlarges as the angle of attack increases. XFOIL predicts the polar in $c_l$ and $c_d$ accurately up to the angle of attack of 20°, which is near the maximum $c_l$. The deviation between static URANS and XFOIL increases as the angle increases, which addresses the limitation of utilizing two-dimensional tools for large WT applications. The influence of the mesh size on the prediction of stall cycles in $c_l$, $c_d$, and the onset of static stall at $\alpha_{ss} = 15°$ is similar. Fig. 3a predicts an earlier onset of deep stall at 23° when the other setups show light stall behavior. The cycle-to-cycle variation is large in this case, which is not physical phenomenon, rather numerical error. Figure 3d slightly under-predicts the variation in $c_l$ during DS, especially the maximum lift and lift during a down-pitching, where DSVs are developed into large-scale three-dimensional vortices. The conventional choice of spanwise extension as similar to or smaller than a chord length might not be sufficient to resolve DS of large WTs. The AR of 2.5 with 20 cells are chosen for the studies in this paper, nevertheless, the influence of the domain should continue to be investigated in the future, focusing on the three-dimensional aspect of DSVs.

Figure 4 shows the dynamic stall cycles from different maximum CFL number cases of 100, 50, and 10. The pitching angle is chosen to be $17° \pm 15°$ to examine the influence of the time-step during a deeper stall cycle compared to the mesh study. The cycle converges towards maximum CFL of 10. This setup results in CFL number of the majority of the cells under 1 at every time step. Maximum CFL number of 5 would require very high computational power. The prediction of $\alpha_{ss} = 15°$ and maximum $c_l$ at $29° - 30°$ is similar for all different CFL numbers. However, the difference is distinct in $c_l$ and $c_d$ near the maximum angle $\alpha = 30 - 32°$ especially during down-pitching, which is relevant estimating LESP and later developing reduced order model. Therefore, the maximum CFL number 10 is chosen for further simulations.

## 4 Features of Dynamic Stall

### 4.1 Sensitivity of Dynamic Stall towards Pitching Angle

The dynamic stall cycles in different mean and pitching angles are shown in Fig. 5. The onset angle of the static stall is approximately $\alpha_{ss} = 15°$ as it was depicted in the previous section. After exceeding $\alpha_{ss} = 15°$, small fluctuations are observed in the dynamic signals of $c_l$ and $c_d$ as the flow becomes unsteady. The fluctuation and the cycle-to-cycle



**Figure 5.** Lift and drag coefficients of different mean and pitching angles. (a) $\bar{\alpha} = 17°$ $\pm 8°$ (left), $\pm 15°$ (right). (a) $\bar{\alpha} = 20°$ $\pm 5°$ (left), $\pm 10°$ (middle), $\pm 15°$ (right).

variation increase further and become distinctly more chaotic around $\alpha_* = 28°$. This is also where the dynamic $c_d$
rapidly changes during down-pitching, and the static lift abruptly decreases due to the flow separation without
reattachment in the LE region, i.e. where an open flow separation is present on the suction side of the profile. The



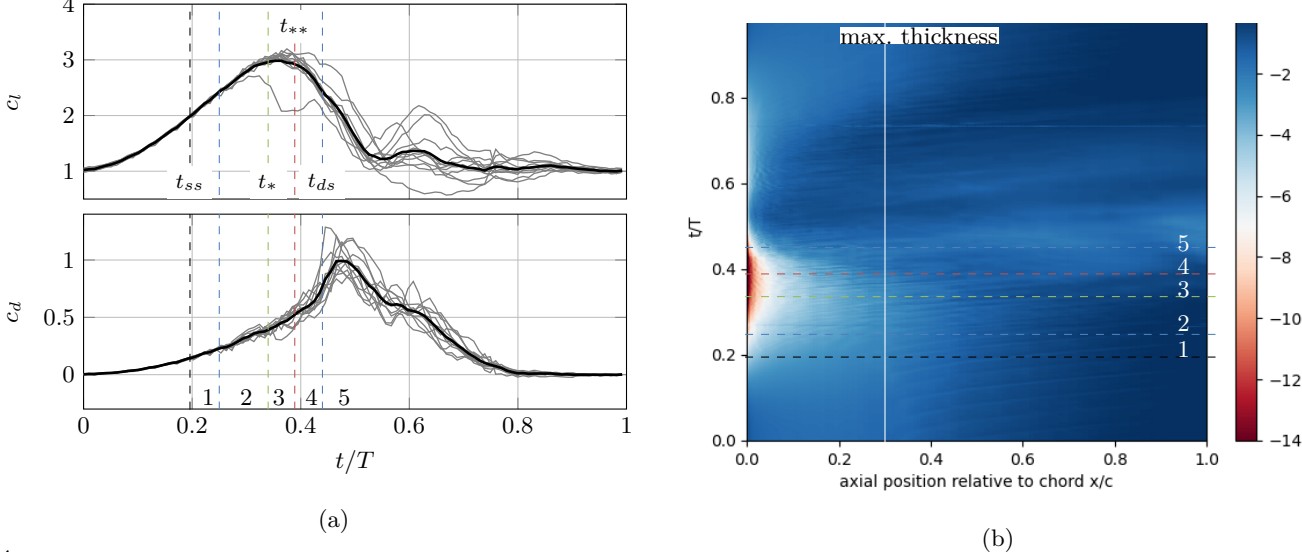

**Figure 6.** Evolution of dynamic stall in different phases, static stall at $t_{ss} = 0.196$, localized stall at $t_* = 0.34$, full stall at $t_{**} = 0.389$, and deep stall at $t_{ds} = 0.45$: (a) Phase-averaged (black) and individual (gray) lift and drag coefficients, (b) Spatiotemporal plot of phase-averaged pressure coefficient $c_p$ on the suction side.

deviation in static polar between URANS and XOIL is large in this angle because XFOIL does not resolve this localized separation behavior. The pitching angle of $20° \pm 15°$ shows a deep stall cycle when the rest investigated in this paper are light stalls. This case experiences a sudden decrease in lift and increase in drag during dynamic stall

at the angle of attack of $\alpha_{**} = 31.5°$, and enters deep stall at $\alpha_{ds} = 34°$. The static polar shows a second drop around $\alpha_{ds} = 34°$, which indicates the complete stall in the LE region. These distinct angles, $\alpha_{ss}$, $\alpha_*$, $\alpha_{**}$, and $\alpha_{ds}$ can be set as criteria for this airfoil design. However, to generalize this observation for different airfoils during a design process and to find the correlation between the airfoil parameters and the critical angles, comprehensive understanding and analysis of the flow field is necessary.

## 4.2    Life Cycle of Dynamic Stall

The life cycle of dynamic stall is conventionally categorized into the primary instability stage and the vortex forming stage during an up-pitching motion. The primary instability stage is where the flow becomes unstable exceeding the static stall limit ($t_{ss}$). The shear layer starts rolling up forming localized vortices. During the vortex forming stage ($t_*$), the shear layer is rolled up together with the leading-edge separation vortex (LEV), forming a large DSV

(Fig. 1a). DSVs are detached after entering this stage, leading the decrease in lift. Dynamic stall of a thick airfoil under very high Reynolds number reveals that this classification is not completely applicable due to the distinct localized characteristics. Figure 6a shows the lift and drag coefficients, and Fig. 6b the pressure coefficient $c_p$ on





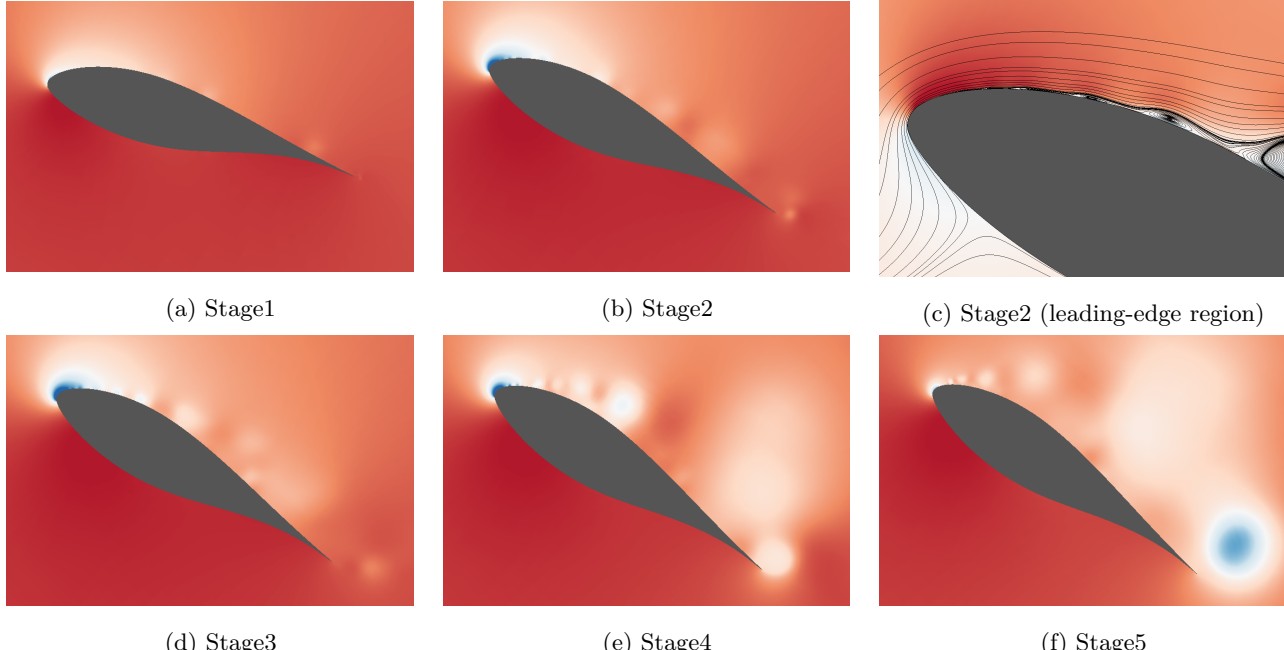

Figure 7. Instantaneous pressure contours through a life cycle of dynamic stall, low pressure level in blue. (a) Stage1: Initial instability, (b) Stage2: Attached vortex street, (c) Streamline near leading-edge (d) Stage3: Lift peak, localized stall, (e) Stage4: Full stall, (f) Stage5: Deep stall

the suction-side of the airfoil with the individual stages marked, while Fig. 7 shows the pressure contours at the mid-span. The definition and description of each evolution phase are as follows:

- Pre-stall: At a low angle of attack, small separations form near $0.3\,c$ where the maximum thickness of the airfoil is found. Those separations are convected downstream while the leading edge area is steady. Those are low intensity and constant vortices, which are not influenced by the pitching angle in this region.

- Stage1 (Initial instability, localized vortex formation): Exceeding the static stall at $t_{ss} = 0.196$, linear increase in the lift and drag is observed. Few localized small-scale vortices form independently in the LE region $(0 - 0.3\,c)$

and mid-chord to TE region $(0.3 - 1\,c)$. Those vortices are low intensity and quickly dissipate while being convected downstream. Their traces can be seen at the spatiotemporal plot in Fig. 6b at $t = 0.196 - 0.25$. The cycle-to-cycle variation is negligible at this initial instability stage, implying a quasi-steady state.

- Stage2 (Dynamic stall vortex formation, attached vortex street): Further increase in the pitch angle results in gaining lift, although the increase is decelerated compared to the previous stage. The pressure contour shows that

the LE is strongly separated, generating small-scale vortices (Fig. 7c). These high intensity LEVs travel downstream, they develop into DSVs as a result of interaction with the mid-chord vortices. It results in multiple vortices of large to small-scales in various intensities, attached along the entire suction-side of the airfoil. The cycle-to-cycle variation is increased compared to the earlier stage.





- Stage3 (Lift plateau, localized stall): Although LE suction still increases and generate stronger vortices at $0-0.3\,c$, the total lift does not increase since the stall is initiated at mid-chord to TE ($0.3-1\,c$). This is apparent in Fig. 6b and Fig. 7d as detached and dissipating vortices. The lift curve $c_l$ shows a plateau from $t_*$. This intermediate stage is a consequence of turbulent flow around LE and abrupt changes of blade thickness along the chamber line. Therefore, this is not observed in dynamic stalls of thin airfoils under low Reynolds numbers, they enter directly into the full stall region. This stage functions as a delay of deep stall and is beneficial to gain further lift during a pitching motion.

- Stage4 (Full stall): LE region gradually stalls after reaching its maximum capacity at $t_{**} = 0.389$. In this stage, the suction-side is fully stalled and the trailing-edge vortex starts appearing (Fig. 7e). This process results in a drastic decrease in lift $c_l$ and increase in drag $c_d$.

- Stage5 (Deep stall, shedding of dynamic stall vortex): The airfoil is fully stalled, lift is decreased and drag is increased continuously. The beginning of this stage, $t_{ds}$ might involve slight increase in lift due to the shedding of the large DSVs.

- Post-stall: Leaving the stall by a down-pitching motion at $t = 0.5$, the lift usually fluctuates and is returned to the pre-stall state, featuring LE reattachment. For the current study, this returning process is slightly delayed since the LEVs are still generated at $t = 0.5 - 0.75$, preventing a rapid decrease in lift. This is related to the characteristics of delayed and stretched stall region during the up-pitching motion.

## 4.3 Vortex Dynamics

Spectral analysis reveals the spatial and temporal evolution of vortices described earlier. Figure 8 shows frequency spectra along the airfoil suction-side in different time steps. The axial position is in the log-scale to highlight the LE region, where the dynamic unsteady flow separation is found. The origin and evolution of small to large-scale DSVs are traced following the life cycle defined in the previous section. As the angle of attack exceeds the static stall limit (Fig. 8a), the increase in unsteady pressure coefficient $c_p$ is detected at $0.3 - 1\,c$ at very low frequency. This is the result of localized separations defined earlier. The pressure gradient along the airfoil surface at this stage is small, so that the vortices are slowly convected downstream and dissipate. The LE region is at this stage still static (dark red contour), the entire airfoil is in a quasi-steady state. As the cycle enters the Stage1 (Fig. 8b), LE separation at $0.025\,c$ is initiated with discrete frequency contents. The local Reynolds number $\mathrm{Re}_x = 3.8e5$, indicates that this is a turbulent separation. The previously separated region at $0.3 - 1\,c$ becomes slightly more unsteady. The vortices originated from leading-edge and mid-chord are separated, persisting their own characteristics, indicating no large-scale DSVs are formed yet. Towards Stage2 (Fig. 8c and 8d), the LE separation point is shifted to further upstream at $0.006\,c$. The highest peak is observed at approximately $\mathrm{St} = 25$, this vortex is then convected downstream with a slightly decreasing frequency. This peak corresponds to the local Reynolds number of $\mathrm{Re}_x = 9e4$ and Strouhal number of $\mathrm{St}_x = 0.15$, indicating the laminar separation bubble and its shedding. Laminar separation is more abrupt compared to the previous turbulent separation. The cascade form of the spectrum can be interpreted as the relative sizes of vortices from LE to TE, the high frequency near LE means small-scale LEVs and low frequency towards




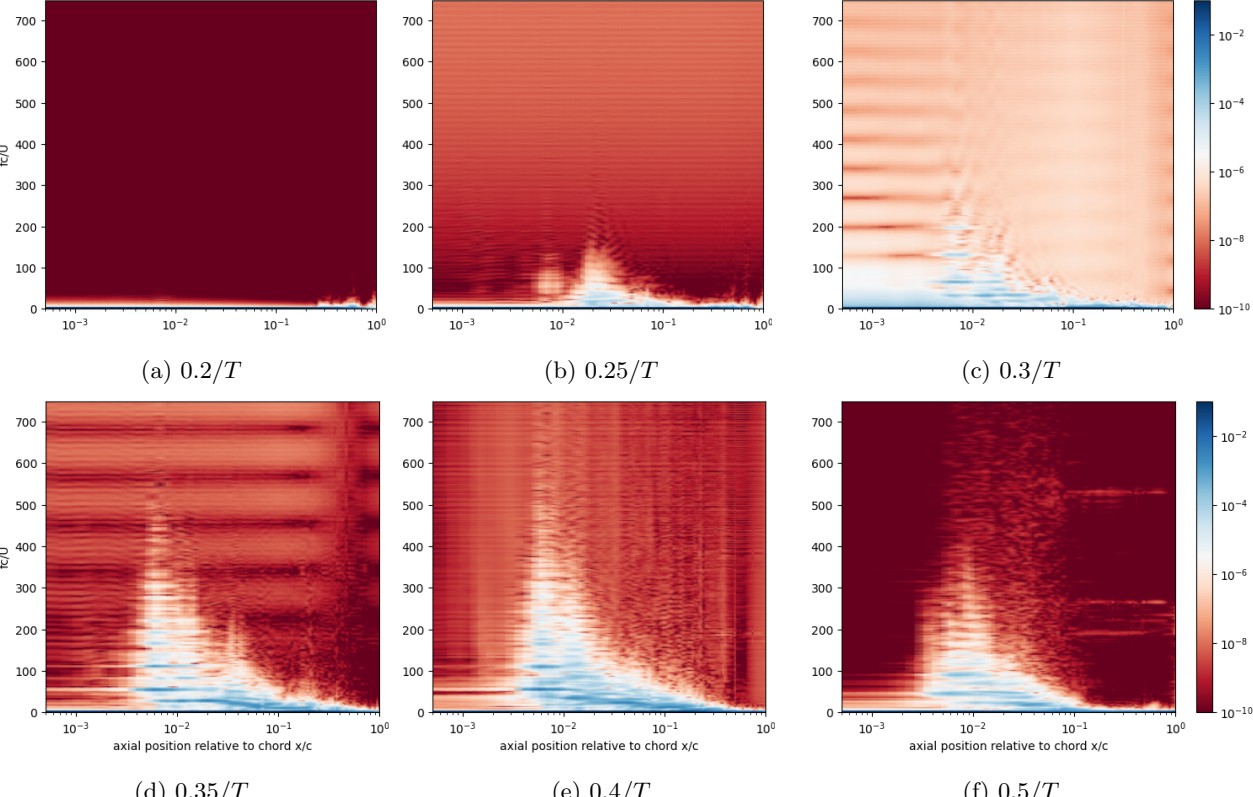

**Figure 8.** Power spectral density of pressure coefficient $c_p$ on the suction side of the airfoil during a dynamic stall, frequency non-dimensionalized as St. (a) Stage1: immediately after onset of static stall, (b) Stage1-2: initial instability, (c) Stage2: attached vortex street, (d) Stage3: maximum lift, (e) Stage4: full stall, (f) Stage5: deep stall

TE means large-scale DSVs. This is also an indication of interaction between the convected LEVs and downstream vortices. DSVs are attached along the entire surface, achieving the maximum lift. At the full-stall of the airfoil (Fig. 8e), the energy is less concentrated in specific frequencies, rather more broad-banded, as a result of dissipation. The DSVs are enlarged and detached on the suction surface. Another noticeable region is the TE, where the flow becomes more unsteady due to the forming of trailing-edge vortex. During the deep stall (Fig. 8f), the unsteady energy is mostly dissipated except in the LE region, where the vortices are still generated.

This analysis not only supports the characterization of the flow around the airfoil, but also can be utilized to design a suppression method of the dynamic stall. The regions that are prone to flow separations can be either actively or passively controlled. Various methods (e.g. surface treatment, air injection, pulse generation) are conventionally applied near the LE to bypass the laminar separation (Visbal and Benton, 2018; De Tavernier et al., 2021). For large WTs, the mid-chord region could be additionally considered for the flow control to delay the localized flow





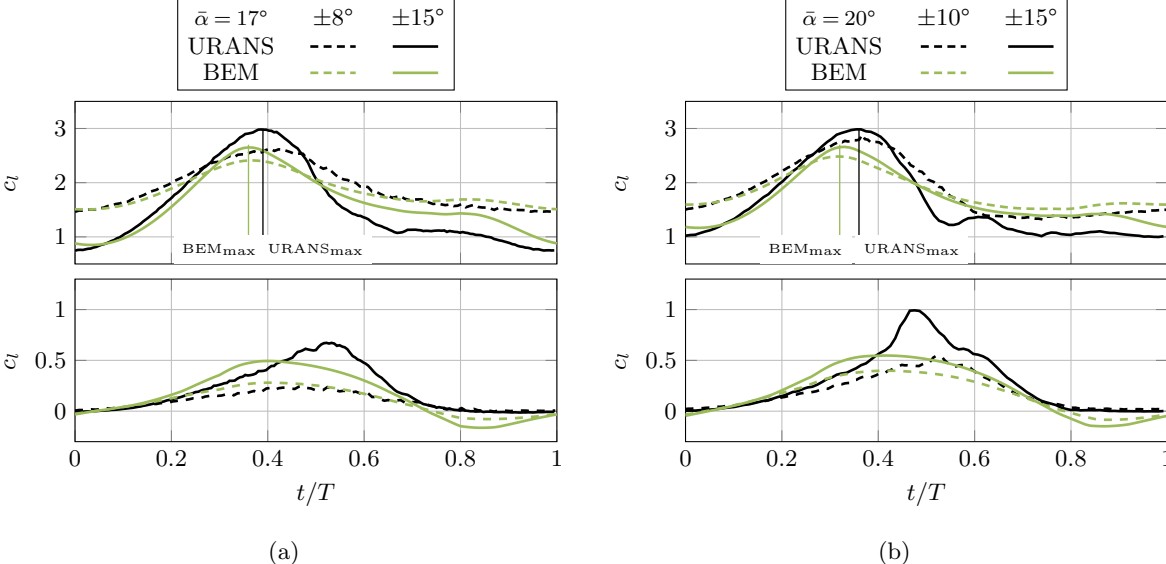

**Figure 9.** Comparison of lift and drag coefficients between phase-averaged URANS and BEM simulations. (a) CaseID 2, (b) CaseID 3 with mark at maximum $c_l$, indicating under-estimation of dynamic stall from BEM

separation. Considering divergent frequency spectra in different axial positions, the method has to be independently
optimized along the airfoil chord.

## 5   Comparison with an Existing BEM Model

The BEM setup is based on static polars provided by the CFD. The airfoil under consideration and the Reynolds
number are the same. The dynamic test cases that are investigated are the same as those listed in Tab. 1. The time
step of the BEM simulation is $\Delta t = 1e^{-2}$ and differs from the time step of the CFD simulation. This time step is
considered adequate for convergence and stability, whereby its reduction has no effect on the results. The exact model
chosen for comparison with the CFD simulations is the 4-states model of Hansen, Gaunaa, and Madsen (known as
the HGM Model), which is a variation of the BLM Model. The HGM Model is analyzed in detail in Branlard et al.
(2022). The unsteady parameters used by the HGM Model to create the hysteresis loops were calculated using the
static polars at Re = 15 M. RANS simulations in combination with XFOIL simulations were used to determine the
static polars for $5° < \alpha < 17°$. The XFOIL polars were utilized for angles up to 10°, while the RANS polars were
employed for angles ranging from $10° < \alpha < 17°$. The static polar data underwent a 3D correction based on the
approach outlined by Du and Selig (1998). Extending the polar curve to cover a range of $-180° < \alpha < 180°$ allows
the determination of the unsteady parameters. The unsteady parameters which are used in the HGM model can
be found in Damiani and Hayman (2019). Other parameters which are set in the HGM Model are the empirical



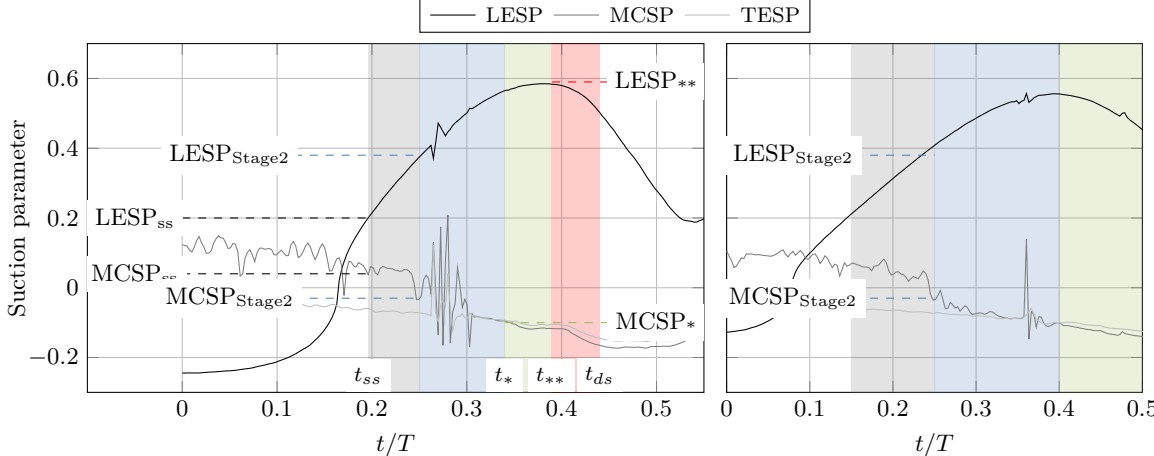

**Figure 10.** Leading-edge suction parameter (LESP), mid-chord suction parameter (MCSP), and trailing-edge suction parameter (TESP) from the pitching case of $20° \pm 15°$ (left) and $20° \pm 10°$ (right). The shaded regions refer to Stage1 (gray): initial instability, Stage2 (blue): attached DSVs, Stage3 (green): maximum lift, Stage4 (red): full stall

determined constants $A_1$, $A_2$, $b_1$, $b_2$, $T_p$, $T_f$, and $T_v$. This paper uses the values calculated by Leishman and Beddoes (1989): $A_1 = 0.3$, $A_2 = 0.7$, $b_1 = 0.14$, $b_2 = 0.53$, $T_p = 1.7$, $T_f = 3$, and $T_v = 6$.

    Figure 9 compares the lift and drag coefficients between URANS and BEM simulations. The BEM result agrees to URANS at the initial vortex formation stage. However, BEM under-predicts the maximum loading by earlier development of dynamic stall. When URANS predicts further increasing $c_l$ due to the delaying effect of dynamic

stall at $t = \text{BEM}_{max}$, BEM enters into dynamic stall. This is reasonable considering the model is optimized for thin airfoils in low Reynolds numbers. The distinct characteristic of Stage2 and Stage3, the strong localization of flow separation and vortex detachment, is not considered in the current BEM model. These two phases should be considered in the future reduced order models like BEM.

## 6   Predicting Onset of Dynamic Stall

While the general feature of each stall phase during a dynamic stall cycle is described earlier, the criteria determining the initialization of each phase is discussed in this section. This can be utilized to develop a reduced-order model in the future. Besides the commonly known leading-edge suction parameter (LESP), which is defined as the chordwise component of airfoil loading at LE $(0 - 0.1 \ c)$, MCSP $(0.3 - 0.4 \ c)$, and TESP $(0.9 - 1 \ c)$ are newly introduced. Conventionally, LESP is sufficient to represent the onset characteristics of each stage of the dynamic stall. Due to

the more detailed stages in this study, MCSP and TESP might be the supporting parameters that fully describe the beginning of individual stages. Figure 10 shows LESP, MCSP, and TESP for two different variations of pitching angle, $20° \pm 15°$ and $20° \pm 10°$. A critical LESP for the onset of static stall (LESP$_{ss}$) is valid as both of the cases





having $\text{LESP}_{\text{ss}} = 0.2$. At the same time, the $\text{MCSP}_{\text{ss}} = 0.04$ and $\text{TESP}_{\text{ss}} = -0.06$ support the indication of onset of static stall, although the sensitivity in these parameters are low. Entering Stage2 at $t = 0.25$, MCSP abruptly

decreases below zero ($\text{MCSP}_{\text{Stage2}} = -0.03$) due to the mid-chord region being under the influence of convected LEV. $\text{LESP}_{\text{Stage2}} = 0.38$ as a criterion is valid as well. Reminding that Stage3 ($t_*$) is characterized by slightly increasing leading-edge suction and complete stall in the rest of the area, MCSP and TESP are better criteria than the LESP. LESP either increases further (case $20°\pm15°$) or decreases (case $20°\pm10°$) depending on the angle setting. $\text{MCSP}_* = 0.1$, $\text{TESP}_* = -0.09$ are found to be valid for both cases. Whether the cycle enters Stage4 (full stall at

$t_{**}$) can be distinguished by all three parameters. However, $\text{LESP}_{**} = 0.59$ must be the determining parameter when $\text{MCSP}_{**}$ and $\text{TESP}_{**}$ are the consequence of the full-stall driven by the stall in the leading edge.

## 7   Conclusions and Outlook

Wind turbines operate in highly unsteady wind conditions, which makes avoiding dynamic stall entirely an impossible task. In order to design airfoils that are less prone to dynamic stall, especially deep stall, a reduced-order model

that can predict the onset of the formation of the dynamic stall vortex is required. The research presented here contributes to the development of such a reduced-order model for wind turbine airfoils of high thickness-to-chord ratio and long chord length by analyzing the unsteady RANS simulations of a pitching FFA-W3-211 airfoil at the Reynolds number of 15 million.

   The dynamic stall cycle is categorized into four phases based on the unsteady vortex dynamics along the suction-

side of the airfoil. The initial instability phase and attached vortex street phase fall into the conventional primary instability stage. Before the occurrence of the deep stall, the peak lift phase, where the loss due to the localized stall is compensated by strengthened leading-edge vortices, is found. Individual stages are characterized in depth from a frequency analysis, where the size and growth of dynamic stall vortices and the interaction between them are presented. The leading-edge suction parameter (LESP), introduced by Ramesh et al. (2014) for predicting the onset

of the vortex formation stage on thin airfoils, was analyzed to test its feasibility for thick airfoils under high Reynolds numbers. It was found that the temporal evolution of the LESP indicates the inception of full stall. Since the flow at the leading-edge remains mostly attached during the dynamic stall cycle, the supplementary parameters which can represent the suction capability at different airfoil locations are needed. For airfoils as they occur on wind turbines, the dynamic stall vortices form and separate initially at mid-chord, which illustrates that monitoring the LESP is

not sufficient for predicting the onset of the initial vortex formation stage for the designer of WTs. Therefore, we introduced the mid-chord suction parameter (MCSP) at $0.3 - 0.4\ c$ and the trailing-edge suction parameter (TESP) at $0.9 - 1\ c$, which are the same suction vector component as LESP, but on locations donwstream of the leading edge. Since the MCSP is based on the pressure at the location of the initial vortex separation, it seems to be a robust supplement for the application to thick airfoils together with the LESP. Analogous to the LESP, the MCSP

and TESP indicate the transition points of the dynamic stall stages.





Future work should investigate the sensitivity of the airfoil camber distribution and the location of the maximum thickness to the stall delay attributed to the localized stall phases. Establishing correlations between $LESP_{crit}$, the temporal evolution of MCSP and TESP, and these airfoil parameters would allow for the development of a reduced-order model that can predict dynamic stall in the design process of new airfoil geometries.

*Author contributions.* HRK conducted the URANS simulations, evaluated the results, and wrote the paper except the parts regarding BEM. JAP conducted the BEM simulation and wrote the methodology and results regarding BEM. JDA wrote the majority of the introduction. HRK, JAD, and JAP worked under the supervision and review of JRS and LW.

*Competing interests.* The authors declare that they have no conflict of interest.

*Acknowledgements.* The authors gratefully acknowledge the computing time granted by the Resource Allocation Board and
provided on the supercomputer Lise and Emmy at NHR@ZIB and NHR@Göttingen as part of the NHR infrastructure. The calculations for this research were conducted with computing resources under the project nii00172. The authors thank the cluster system at the Leibniz Universität of Hannover for the HPC resources, which have contributed to the development of the research results presented here. The present work has been carried out in the subproject A02 within the Collaborative Research Center (CRC) 1463 "Integrated design and operation methodology for offshore megastructures" which is funded by
the Deutsche Forschungsgemeinschaft (DFG, German Research Foundation). The authors thank the DFG for the support.





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
