# Peer review of "Characterization of Dynamic Stall on Wind Turbine Airfoil under High Reynolds Number"

_Wind Energy Science, 2024_

## Author Response (AR1)

Dear the editor and reviewers,

We would like to appreciate your honorable work on organizing the journal and the thoughtful comments based on your experiences and insights. The comments help us to improve the manuscript.

Based on your comments, the manuscript has been revised. Additional response to your questions and comments are found in the next pages, together with the line numbers of the revised manuscript. For the convenience, the line numbers correspond to the track-change file, which is WES-2024-31_trackChange.pdf, not the manuscript itself.

We would like to address two specific questions, one regarding the change of the title, the other the possibility of the nomenclature. The current title is already changed based on the comment from the Reviewer2. The nomenclature is suggested by the Reviewer1, however the template from WES does not equip that. We would like to switch the section 2.2 into the nomenclature if that is allowed.

Sincerely

Hye Rim Kim on the behalf of the authors

**Reviewer1**

A. With the amount of information given in this paper, I do not think I would be able to define the specific angles ($\alpha_{ss}$, $\alpha_*$, $\alpha_{**}$ and $\alpha_{tds}$) on another wind turbine profile. For example the static stall is not clear on a thick airfoil. How do the author define a static stall here? Is it a complete stall of the airfoil? For example Braud et al (Study of the wall pressure variations on the stall inception of a thick cambered profile at high Reynolds number, Physical Review Fluids, 2024) have shown in their recent paper that they did not find a complete stall of the airfoil on a wind turbine profile below 25°. (They also highlighted the importance of 3D effects at high Reynolds numbers).

The definition of $\alpha_*$ as defined by Mulleners and Raffel (The onset of dynamic stall revisited, Exp Fluids, 2012), are based on POD modes on the vorticity field. It seems that here all these angles lack clear definitions.

*Static stall angle at 15° in this case is defined as the lowest angle where the dynamic curve departs from the static curve. This is a common definition for low Reynolds number applications since they regard the leading-edge stall. As you mentioned, in high Reynolds numbers, especially for thick airfoils, there is no complete stall found in this angle of attack. At 15°, the trailing-edge region starts partially stalling, resulting in increase in lift relative to the static curve. In our case, the airfoil is completely stalled at the angle of attack of approximately 35°. We don't suggest a precise method to provide the criteria for those states in this work. The only criteria can be found is critical LESP, MCSP, and TESP at the end of the paper. This study focuses on the characterization of dynamic stall in very high Reynolds number, and provides a first estimation based on the characteristics of lift curse, drag curve, suction parameters, vortex structures, and spectra. POD could be a very good method to set the criteria for those characteristic stages. We would consider that for the further studies when we have more parameter studies completed in this application.*

*The revised parts can be found in line 279-286.*

B. Similarly, I would not be able to compute the Middle Chord Suction Parameter (MCSP) or (TESP) Trailing Edge Suction Parameter on another wind turbine profile. Whereas the Leading-Edge Suction Parameter (LESP) defined by Ramesh has a physical definition (it is based on the first Fourier term in thin-airfoil theory), I do not see a physical sense to the new MCSP and TESP. I did not understand how they were calculated. I do not think they can be defined in a similar way to the LESP using the inviscid flow theory.

*In this numerical study, LESP is calculated as the chordwise projection of leading-edge suction vector at the first 10% of the leading edge as suggested in Deparday and Mulleners (2019) for the evaluation of their experimental data. The MCSP and TESP are calculated in the same manner at 30-40% and 90-100% of the chord, respectively. This is applicable for other profiles as well. As you mentioned, the analytical LESP is defined as the first Fourier coefficient of the Fourier series since it represents the suction peak at the leading-edge. LESP can be calculated as well based on the pressure distribution obtained by vortex-lattice method. MCSP and TESP can be calculated in the same manner, which one of our partner is currently developing focusing on large wind turbine airfoils.*

*The revised parts can be found in line 96-97.*

C. The outcomes of this paper are intended for wind turbines. But I am not sure that the cases studied here are relevant to wind turbines. The applied turbulence intensity is 0.01%, while in general it is at least 8-10%. What would be the operating angle of attack on such a profile, and what would the expected variations in angle of attack be? For example the sinusoidal motion 20°+-15° does not seem realistic to me. Some contextualisation may help to appreciate the importance of dynamic stall on a wind turbine blade section.

*Considering the large wind turbine encountering laminar flow at the high altitude, we have initiated our studies with very low turbulence intensity. The future studies will be conducted with different turbulence intensities considering the atmospheric boundary conditions and the wakes of other wind turbines in the wind farm. The sinusoidal motion could be still applicable for wind turbine considering a quasi-steady rotor oscillation induced by fluid-structure interaction. This can present in low angle of attack region with a moderate oscillating amplitude. 20°+/-15° is a very unlike kinematic condition in real wind turbines. However, it is still necessary to understand the characteristics of dynamic still in these extreme cases to develop a comprehensive dynamic stall model in the future.*

*The revised parts can be found in line 135-137, 221-224, and 461-462.*

I have other minor comments:

1. end of page 6: "the conclusions regarding the formation of dynamic stall on future large WTs remain the same". I am not sure to what the authors refer to when they write "the same".

*This is revised in line 226-227.*

2. page 5: LE, TE, and CFL (and BLM and k in page 6) are used but were not introduced before. I do not know what CFL means.

*The abbreviations are more thoroughly included. This is revised in line 79, 156, 299, 320 and other parts as well. CFL is Courant-Friedrichs-Lewy number, which judges the convergence of partial differential equations.*

3. section 2.2 seems more a section for the introduction. The beginning of section 5 may be more appropriate in this section "2. Methodology".

*The parts are shifted so it fits better in each section. This is revised in line 65-84, 229-240, and 390-404.*

4. Section 2.4: This section might be better used as a nomenclature (if the WES template allows it).

*I agree on that, I have addressed the possibility to the editor.*

5. Figure 3: It is hard to compare the different plots in figure 3. It may be easier to compare them if they are on the same plot (and probably with the time as x-axis). The phase-average value would probably suffice here to compare the general evolution of the numbers of cell or maximum CFL.

*The phase-averaged value might not be sufficient. To distinguish the pure numerical cycle-to-cycle variation (e.g. Fig.3(a)) to physical cycle-to-cycle-variation, we have decided to show the figures individually.*

6. Why is the pitching case different for the mesh and time-step studies? (17°+-8° for the mesh study and 17°+-15° for the time-step study). The same pitching case for both studies would probably ease the comparison.

*Considering that mesh-studies are normally conducted in static conditions, 17°+/-8° condition is more than enough for the mesh-study. In this case, repeating the mesh-study for 17°+/-15° would be unpractical since the simulation requires a huge amount of the computational power. The time-step study is however important for predicting dynamic performance. After the mesh-study, we have decided to conduct the time-step study for a deeper dynamic stall cycle, so at 17°+/-15°. Since those two studies are completely independent, we would keep it as it is. I hope this is fine for you and the readers.*

7. Figure 4. What do the arrows mean in figure 4?

*They were pointing the highlights, but we have removed to avoid confusion. This is revised in Fig. 4.*

8. page 10. What do the author mean with the term "open flow separation"?

*This is a kind of idiom to describe flow separation without reattachment. For the clarity, this is revised in line 292-294.*

9. Figure 6: The angle of attack as x-axis (place on the top of the plot for example) would help to visualise the time and the angle of attack at the same time.

10. Figure 6: Please indicate $t_{ss}$, $t_*$, $t_{**}$ and $t_{ds}$ in the x-axis of figure 6 and not just in the legend. It is harder to follow without these specific times in the graph.

11. Figure 6: the colorbar probably represents the pressure coefficient. Could you please mention it on top of the colorbar?

12. Figure 6: The colormap used is divergent, with a white color in the middle which "separates" the blue and red color. But the white value has no signifiant value here. A convergent colormap might be more appropriate here to better visualise the transitions in the pressure coefficients.

13. Figure 6: A Cp of -14 seems a lot to me. I cannot recall such a high absolute value even in simulations. Could the author confirm this extremum please?

*Figure 6 is revised according to the comments #9-#13.*

14. Figure 7: The instantaneous pressure contours are probably not the best to visualise the vortices described in the text. The z-vorticity contours or the Q-criterion might be more appropriate.

*We have originally tested the contours of Q-criterion or z-vorticity, however it doesn't display the vortices better than Cp-distribution because of the exponential nature of the distribution. Therefore, we have decided to keep the contours of the pressure coefficient. The pressure sinks in Cp contours are very visible and the vortices are supported by Fig.7(c)*

*The example contour of the spanwise vorticity at t=0.32/T is attached here.*

[Figure]

15. Figure 7: Could the author add a colorbar here for the pressure value. It seems to be a different scale to the colorbar shown in figure 6, which uses the same colormap.

16. Figure 7: For each subcaption, it would be good to add the angle of attack and time, when these snapshot were taken, and if they correspond to a salient time such as $t_{ss}$, $t_*$, $t_{**}$ or $t_{ds}$.

*Figure 7 is revised according to the comments #15-#16.*

17. Page 13: What do the authors consider to be a thin profile and low Reynolds numbers? For example in a sinusoidal pitching airfoil cited in this paper (Deparday and Mulleners, PoF, 2019) or Deparday et al, JFM, 2022 (Experimental quantification of unsteady leading-edge flow separation), it seems there is a similar plateau of Cl but the airfoil is thinner and the Reynolds number lower, which would contradict the conclusions here.

*Thick airfoil is defined as (t/c>0.21) as it is written in the abstract. More insight regarding plateau of cl is revised in line 330-339,*

18. Figure 8: I have the same comments about the divergent colormap, and no mention about what the colorbar represents.

*Figure 8 is revised.*

19. Figure 8: It seems there is a periodic pattern with the Strouhal number. Could the author confirm this is not an artifact due to the time step of the simulations?
*The setup is additionally described in line 356-360. We can confirm that this is not artifact.*

20. I do not understand why BEM is applied here. Did the authors model a rotating wind turbine blade? What is the geometry of the blade then, the rotational speed?

*The conditions of BEM (HGM model) simulation are exactly the same as the CFD setup. We would like to emphasize here that the models calibrated for thin airfoil and low Reynolds application is not able to predict the dynamic stall cycle of our application because the characteristics of the dynamic stall are different.*

21. I would like to mention to the authors a new model for dynamic stall recently published (Bangga et al, Development and Validation of the IAG Dynamic Stall Model in State-Space Representation for Wind Turbine Airfoils, 2023, https://doi.org/10.3390/en16103994)

*Thank you very much for the suggestion. The manuscript is revised in line 80-83 and 407-408, mentioning this model and we would like to apply the model for the future studies.*

**Reviewer2**

1.   First, the title is misleading. The paper is focused on wind turbine airfoils, not wind turbine itself. In my opinion it is not appropriate to label it as "wind turbines". The dynamic stall behavior in wind turbines is much more complex than just pitching airfoils because it also involves strong flexibilities, instability, plunging, heave and all complex 3D flow field. Our group luckily had a chance to study that
in:  https://iopscience.iop.org/article/10.1088/1742-6596/2626/1/012026/meta

Just as an example, but you really could find the characteristics in lots other papers. As far as I can see, the present paper only discusses the aspects on airfoils. For sure, please correct me if I read your paper in a completely wrong manner :)

*We have suggested the new title and addressed this question to the editor.*

2.   Minor but could be helpful: I would suggest adding a table of symbols, honestly I had difficulty finding out the meaning of alpha* :)

*The manuscript is revised highlighting the definitions in line 280-281 and 313-314, and we have tried to defined this more clear throughout the manuscript.*

3.   Why did you use fully turbulent solutions? Any solid reasoning not to use transitional model or to enforce transition at certain locations (like how measurement is usually done)? I think that the argument that high Re will have a fully turbulent flow is a bit strong opinion, since it will still have transition regime to a certain degree, and it will impact the stall behavior.

*Kiefer et. al. (2022) investigated different Reynolds number of 0.5\*10^6, 2.0\*10^6, and 5.0\*10^6. They found out that for Re=5.0\*10^6, the boundary layer is transitioned to turbulence, upstream of the laminar separation point of low Reynolds numbers. That is why we first applied the fully turbulent setup. We would like to apply transition model for our future studies to make sure our assumption. The manuscript is revised in 144-147.*

4.   I think the normalization for lift is not right, are we missing "0.5" factor? Or is it intended not to have the usual formulation?

*Yes, the factor 0.5 is missing, this corrected in line 210.*

5.   You compared URANS with XFOIL, by default XFOIL includes transition modelling, how did you align them?

*This might support our argument to apply fully turbulent model. XFOIL scripts that free*

*transition occurs at x/c=0.002 at AoA=20°, which is very near to the leading edge. The manuscript is revised in line 268-260.*

6.  I agree with the other reviewer that you used the term "BEM" for the engineering model calculations, it is more appropriate to label it as HGM model. This model was developed by Hansen, Gauna and Madsen which was based on the Beddoes-Leishman (BL) dynamic stall model. In short, theoretically this is an incompressible version of the BL dynamic stall model. I think it is not fair not to cite the original author (https://orbit.dtu.dk/files/7711084/ris_r_1354.pdf).

*The term is corrected to either BL or HGM model throughout the manuscript, and the original publication is now cited, such as in line 78, 229-240, and 405-4413.*

7.  I thank the other reviewer for bringing up our recent work in dynamic stall modelling. We recently developed the IAG dynamic stall model for wind turbine airfoils and have tested it against experimental data of pitching airfoils at various conditions.

You might observe in our paper that the drawback of underestimating the peaks of the loads for the incompressible BL model (which you observe in your paper) is better solved when using the IAG model.

The papers are here:

https://wes.copernicus.org/articles/5/1037/2020/

and

https://www.mdpi.com/1996-1073/16/10/3994

and we have tested the model against BL model on 3 different wind turbines under design load cases in the recent Torque conference. It is not yet published but I can send the paper to you if you are interested to read.

*Thank you very much for the suggestion. The manuscript is revised in line 80-83 and 407-408, mentioning this model and we would like to apply the model for the future studies.*

---

## Author Response (AR2)

Dear the editor and reviewers,

We would like to appreciate your honorable work on organizing the journal and the thoughtful comments based on your experiences and insights. The comments help us again to improve the manuscript.

Based on your comments, the manuscript has been revised. We tried to justify the use of URANS predicting dynamic stall and XFOIL for the polar prediction, still mentioning the limitations of using those more specifically. The definitions and changes of MCSP and TESP are more specifically described. The changes are marked as orange in the track-change file.

Sincerely

Hye Rim Kim on the behalf of the authors

---

## Author Response (AR3)

Dear the editor and reviewers,

Thank you again for the final review to support us to publish the paper. Equation 7 and Figure 10 have been reworked, following the comment. Another proofreading has been done to correct typos throughout the whole manuscript.

Sincerely

Hye Rim Kim on the behalf of the authors